# A Comparison of Meeting Physical Activity and Screen Time Recommendations between Canadian Youth Living in Rural and Urban Communities: A Nationally Representative Cross-Sectional Analysis

**DOI:** 10.3390/ijerph19074394

**Published:** 2022-04-06

**Authors:** Taru Manyanga, Chelsea Pelletier, Stephanie A. Prince, Eun-Young Lee, Larine Sluggett, Justin J. Lang

**Affiliations:** 1Division of Medical Sciences, University of Northern British Columbia, Prince George, BC V2N 4Z9, Canada; 2Department of Physical Therapy, Faculty of Medicine, University of British Columbia, Vancouver, BC V6T 1Z3, Canada; 3Faculty of Human and Health Sciences, School of Health Sciences, University of Northern British Columbia, Prince George, BC V2N 4Z9, Canada; chelsea.pelletier@unbc.ca; 4Centre for Surveillance and Applied Research, Public Health Agency of Canada, Ottawa, ON K1A 0K9, Canada; stephanie.prince.ware@phac-aspc.gc.ca (S.A.P.); justin.lang@phac-aspc.gc.ca (J.J.L.); 5School of Epidemiology and Public Health, Faculty of Medicine, University of Ottawa, Ottawa, ON K1G 5Z3, Canada; 6School of Kinesiology & Health Studies, Queen’s University, Kingston, ON K7L 3N6, Canada; eunyoung.lee@queensu.ca; 7Northern Medical Program, Division of Medical Sciences, University of Northern British Columbia, Prince George, BC V2N 4Z9, Canada; larine.sluggett@unbc.ca; 8Department of Mathematics and Statistics, Faculty of Science, Carleton University, Ottawa, ON K1S 5B6, Canada

**Keywords:** urban and rural comparison, Canadian youth, physical activity and recreational screen time recommendations, sex/gender differences in PA

## Abstract

Meeting the physical activity (PA) and recreational screen time recommendations for children and young people is associated with several health benefits. The purpose of this study was to compare the odds of meeting PA and recreational screen time recommendations between the Canadian youth living in urban versus rural communities. We analyzed nationally representative cross-sectional data collected as part of the 2017–2018 cycles of the Canadian Community Health Survey among young people aged 12–17 years. PA and screen time were self-reported. Sex-specific multivariable logistic regression models were used to estimate the odds of meeting individual and combined PA and recreational screen time recommendations by rural and urban status after adjusting for individual, socioeconomic, and seasonal covariates. The odds of meeting the PA recommendation were not statistically significantly different among males (OR = 1.01, 95% CI: 0.86–1.18) or females (OR 1.05, 95% CI: 0.99–1.11) living in urban versus rural communities. The odds of meeting the recreational screen time recommendations were statistically significantly lower among male (OR = 0.71, 95% CI: 0.65–0.77) and female (OR = 0.71, 95% CI: 0.59–0.86) youth living in urban compared to those in rural communities. The odds of meeting the combined PA and screen time recommendations were statistically significantly lower among urban males (OR = 0.75, 95% CI: 0.71–0.81) but not females (OR = 0.82, 95% CI: 0.58–1.15) than those from rural communities. These findings suggest that residential context (i.e., urban versus rural) may have a differential impact on meeting the combined PA and screen time recommendations among the male and female Canadian youth. Future research should investigate these differences using device-based measures.

## 1. Introduction

The Canadian 24-Hour Movement Guidelines [1] recommend that, in a single day, children and young people aged 5–17 years should accumulate at least an average of 60 min of moderate-to-vigorous intensity physical activity (MVPA) and no more than 2 h of recreational screen time (hereafter referred to as screen time). For children and young people, meeting the PA and screen time recommendations is associated with improved physical, mental and cognitive health outcomes [2,3]. Despite these health benefits, surveillance data consistently show concerning levels of insufficient PA and high screen time among the young [4]. For example, device-measured PA and self-reported screen time data from the 2014–2015 Canadian Health Measures Survey (CHMS) showed that 37.6% and 28.5% of Canadian children and youth met the PA and screen time recommendations, respectively [4]. The most current device-measured PA data from the 2016–2017 cycle of the CHMS show that 39.2% of Canadian children and youth met the PA recommendation. Meanwhile, self-reported data from the 2016–2017 and 2018–2019 cycles of the CHMS indicate that 59.9% and 53.3% of Canadian children and youth met the screen time recommendation, respectively [4].

A paper led by the World Health Organization (WHO) recently reported that participation in PA differs by region and geographical location [5], suggesting that living in rural and urban settings may provide different opportunities for youth to meet the PA and screen time recommendations. For example, in Canada, rural areas have been found to have lower walkability scores compared to urban areas and exposure to high walkable neighborhoods was associated with utilitarian walking [6]. Similarly, in Scotland, McCrorie et al. (2020) [7] found that rural areas were significantly less walkable compared to urban areas. Although it is well established that active living environments differ between urban and rural settings [8], the impact on PA and screen time among youth is largely equivocal. Some studies [9,10,11] have reported higher PA among rural compared to urban youth, while others [12] showed higher MVPA only among rural girls but not boys compared to their urban counterparts. Others [13] showed no differences in PA levels between rural and urban children and youth. Still others [14,15] have reported lower PA levels among rural compared to urban youth dwellers during winter but not summer. 

In Canada, self-reported data from earlier cycles of the Canadian Community Health Survey (CCHS), showed only rural-dwelling girls but not boys were more likely to meet the PA recommendation compared to those from urban communities [16]. Harvey et al. (2017) [17] showed that Canadian children attending schools in larger urban settings were more likely to meet the PA recommendation than those attending schools in smaller urban or rural settings, whereas device-measured data by Nyström et al. (2019) [18] indicated no significant differences in MVPA between urban and rural Canadian children. Although device-based, the data from Nyström et al. (2019) [18] were collected in only three regions of Canada, Nadeau et al. (2016) reported on older (2003–2012) cycles of the CCHS and only compared PA but not screen time between Canadian youth living in urban versus rural communities. Moreover, these Canadian studies [16,17,18] assessed meeting the PA recommendation as the accumulation of at least 60 min of MVPA on six or all seven days of the week, as opposed to averaging the daily MVPA across the week which is currently used for surveillance purposes [19].

Furthermore, current evidence [20] and trends suggest that the prevalence of insufficient PA and high screen time among youth may, at best be stable or increasing [21,22,23] over time, thus, it is important to understand ifs the previously observed differences in self-reported PA adherence between urban and rural youth remain or have increased. Using data from the 2017–2018 CCHS, this study compared the odds of meeting independent and combined PA and screen time recommendations between Canadian youth aged 12 to 17 years living in urban versus rural communities. In preparing this manuscript, we followed the STROBE guidelines for cross-sectional studies (Appendix A).

## 2. Methods

### 2.1. Data Source 

Data for this study were obtained from the 2017 and 2018 cycles of the CCHS annual component. The CCHS [24] is a nationally representative, cross-sectional survey that uses computer-assisted interview-administered questionnaires to collect self-reported information related to the health status, health care utilization and health determinants of the Canadian population [24]. The CCHS covers persons who are 12 years and older, living in Canada’s ten provinces and three territories. It excludes persons living on reserves and other Aboriginal settlements; full-time members of the Canadian Forces; the institutionalized population, children aged 12–17 years who live in foster care, and persons living in the Quebec health regions of Région des Terres-Cries-de-la-Baie-James. These exclusions represent less than 3% of the eligible age groups. More details about the survey’s instrument design, sampling methods, variable coding and weighting are reported elsewhere [24]. 

### 2.2. Study Sample 

After excluding data from adults (i.e., ≥18 years old; *n* = 96,603), we included Canadian youth between the ages of 12 and 17 years, who participated in the 2017 and 2018 cycles of the CCHS (*n* = 7962). We excluded 1786 youth respondents with missing PA (*n* = 1243) and screen time (*n* = 543) data. The final sample size after all exclusions was 6176. Respondents with missing PA and screen time data were not statistically significantly different from those with complete data in BMI z-scores, sex, or perceived mental health (Appendix A). There were statistically significantly more 12–13 and fewer 16–17-year-old respondents with missing data than those with complete data. Additionally, a statistically significantly higher proportion of respondents with complete data reported having very good or excellent general health compared to those with missing data (Appendix A).

### 2.3. Variables

#### 2.3.1. Meeting the PA Recommendation (Dependent Variable)

In the CCHS, regardless of year of collection, levels of PA were self-reported. All respondents were asked PA-related questions [24] that assessed participation, frequency, duration and intensity in the last seven days. For example, respondents were asked if they used active transportation, or participated in recreational PA including sporting activities, and PA related to domestic chores or paid/unpaid work in the last seven days. To conform with the current recommendation [1], self-reported PA was dichotomized into those who obtained a daily average equal to or exceeding 60 min of MVPA (meeting recommendation) compared to those who did not. Population level PA estimates from the CCHS have previously been shown to have overall good agreement with device-measured data [25].

#### 2.3.2. Meeting the Screen Time Recommendation (Dependent Variable)

CCHS respondents were asked two questions specific to screen time [24] in the last seven days: on a school or work day, how much of their free time did they spend watching television or a screen on any electronic device while sitting or lying down; and on a day that was not a school or workday, how much of their free time did they spend watching television or a screen on any electronic device while sitting or lying down? For both questions, response options included: less than 2 h per day; more than 2 h but less than 4 h; 4 h to less than 6 h; 6 h to less than 8 h; 8 h or more per day. The midpoint value (1, 3, 5, 7, 9 h) for each categorical response was used to recode responses into a daily continuous value of hours per day of screen time. A weighted daily average screen time score was generated as follows: ((hours of screen time on weekdays × 5) + (hours of screen time on weekend days × 2))/7. An average of 2 h or less per day was required to meet the screen time recommendation. 

#### 2.3.3. Rural or Urban Location (Independent Variable)

We used Statistics Canada’s definition of rural and urban communities [26]. Respondents were identified as living inside a population centre (urban area) or a rural area. A population centre was described as continuously built-up areas, with a population concentration of 1000 or more and a population density of 400 residents or more per square kilometer. A rural community was defined as all territory lying outside population centers, including small towns, villages and other places with population concentration of less than 1000 people. A rural area could either be within or outside of a census metropolitan area (CMA) or census agglomeration (CA) [26].

#### 2.3.4. Covariates

Age was categorized into three groups: 12–13, 14–15, and 16–17 years. CCHS respondents were asked to identify their biological sex at birth as male or female. Respondents were categorized as white or non-white/racial minority (indigenous, and other visible minority groups). Respondents were asked to rate their general and mental health as: ‘poor’, ‘fair’, ‘good’, ‘very good’ or ‘excellent’. Annual household income was included using a five-category variable (≤CAD 29,999; 30,000–59,999; 60,000–99,999; 100,000–149,999; ≥150,000). Self-reported height and weight were used to derive body mass index (BMI) z-scores using the WHO classification [27] and included four BMI categories (≤−2 = thinness, −2 to +1 = normal weight, +1 to 2 = overweight, >2 = obesity). Season of data collection was also included (January–March; April–June; July–September; October–December).

### 2.4. Statistical Analyses

Analyses were conducted using SAS Enterprise Guide 7.1. To account for the sample design of the CCHS, bootstrap procedures were applied to calculate 95% confidence intervals. Sample weights were applied to account for the complex survey design and to minimize the impact of non-response bias. Logistic regression (proc surveylogistic) was used to calculate the odds of meeting PA and screen time recommendations based on rural or urban residence. Living in rural areas was treated as the referent category for all analyses. We used a model-building approach that added individual, socioeconomic, and seasonal covariates based on theoretical knowledge of potential confounding variables as has been done previously [16]. Interaction terms between location (urban versus rural) and sex were used to assess whether the associations between location and the odds of meeting the PA, screen time or both recommendations differed by sex. Given the known differences in PA participation [28,29,30] and screen time [31,32] between males and females, results from multivariable logistic regression models for the odds of meeting PA, screen time and the combined recommendations are presented separately by sex. For descriptive statistics, statistically significant differences between the urban and rural respondents were identified if the 95% confidence intervals did not overlap. This 95% confidence limit approach is conservative as statistically significant differences at *p* < 0.05 could occur when confidence limits slightly overlap. Values were rounded following standard procedures, and as a result, may not add to the total sums [24]. For the logistic regression analysis, statistical significance level was defined as *p* < 0.05. Findings from the fully adjusted models were used as the primary means of interpreting results for this study. 

## 3. Results 

The survey-weighted demographic data for respondents (48.8% female) by rural (*n* = 1786) and urban (*n* = 4390) location are presented in Table 1. More than half (59.1%, 95% CI: 58.4–59.7) of all respondents self-reported meeting the PA recommendation while only 33.9% (95% CI: 32.8–35.0) met the screen time recommendation. We found statistically significant differences between males and females who met the PA (males: 64.3%, 95% CI: 63.4–65.2 vs. females: 53.6%, 95% CI: 51.4–55.7) and screen time (males: 30.7%, 95% CI: 27.6–33.8, females: 37.2%, 95% CI: 36.2–38.2) recommendations. There were statistically significant differences between the proportion of youth living in urban communities (32.1%, 95% CI: 30.2–34.0) and their rural counterparts (41.4%, 95% CI: 40.1–42.8) who self-reported meeting the screen time recommendation. In addition, there were statistically significantly more rural (26.9%, 95% CI: 26.5–27.3) than urban (22.0%, 95% CI: 19.5–24.5) respondents who met the combined PA and screen time recommendations. We also found statistically significantly lower proportions of respondents from urban (7.4%, 95% CI: 6.3–8.5) than rural (9.9%, 95% CI: 9.4–10.5) communities to be categorized as ‘obese’. Most respondents self-reported ‘very good’ or ‘excellent’ general and mental health.

Table 2 presents results from the logistic regression models estimating the odds of meeting the PA recommendation among Canadian youth living in urban versus rural communities. The addition of covariates to the models had little impact on attenuating the corresponding effect sizes. After adjusting for all covariates, our findings show no statistically significant differences in the odds of meeting the daily PA recommendation for males (OR = 1.01, 95% CI: 0.86–1.18) and females (OR = 1.05, 95% CI: 0.99–1.11) living in urban compared to rural communities.

Table 3 presents results from logistic regression models estimating the odds of meeting the screen time recommendation between Canadian youth living in urban versus those in rural communities by sex. The results from the fully adjusted models indicate that males (OR = 0.71, 95% CI: 0.65–0.77) and females (OR = 0.71, 95% CI: 0.59–0.86) living in urban communities had statistically significantly lower odds of meeting the screen time recommendation compared to their rural counterparts.

Table 4 summarizes the odds of meeting the combined PA and screen time recommendations between Canadian youth living in urban compared to rural communities. A fully-adjusted logistic regression model revealed that urban males (OR = 0.75, 95% CI: 0.71–0.81) had statistically significantly lower odds of meeting the combined PA and screen time recommendations than those living in rural communities.

## 4. Discussion

In this study, we used a nationally representative sample of Canadian youth to compare the prevalence and odds of meeting the independent and combined PA and screen time recommendations between those living in urban versus rural communities. More males than females reported meeting the PA recommendation, whereas more females reported meeting the screen time recommendation. There were no statistically significant differences in the prevalence of meeting the PA recommendation between youth living in urban (59.1%, 95% CI: 58.7–59.6) compared to those in rural (58.7%, 95% CI: 57.2–60.2) communities. We found a statistically significantly lower proportion of urban (32.1%, 95% CI: 30.2–34.0) than rural (41.4%, 95% CI: 40.1–42.8) youth who met the screen time recommendation. Covariate-adjusted logistic regression models revealed that rural males, but not females, had statistically significantly higher odds of meeting the combined PA and screen time recommendations than males living in urban communities. Both males and females living in urban communities had lower odds of meeting the screen time recommendation compared to their rural counterparts. There were no statistically significant differences in the odds of meeting the PA recommendation between youth living in urban versus those in rural communities.

In general, data comparing the prevalence of meeting the PA recommendation between the youth living in urban and rural communities are largely limited and inconsistent. A previous narrative review [14] which included studies comparing PA and diet between urban and rural children and youth reported equivocal results. Of the 16 studies evaluating the percentages of meeting the PA recommendation by urban–rural status, three showed no difference (one was specific to boys and not girls), nine indicated that urban youth were less physically active than their rural counterparts (one was specific to girls and not boys), one showed higher MVPA in urban than rural youth while one showed higher MVPA and lower screen (video, computer games) times among rural compared to urban males. Self-reported data in the US [11] found rural youth to be more physically active than their urban counterparts.

In Canada, self-reported data [17] showed higher odds of meeting the PA recommendation among urban compared to rural school children, while device-based data [18] indicated no differences between urban and rural children. Meanwhile, using CCHS data, Nadeau et al. (2016) found higher odds of meeting the PA recommendation among rural females but not males compared to their urban counterparts. Similar to findings from previous reviews [13,14] and single study data [7,8,18], we found no urban–rural differences in the prevalence of meeting the PA recommendation. The inconsistent findings may be related to differences in measurement instruments (self-reported versus device-measured) for PA, the sex composition of samples, domains of PA measured (school, domestic, recreational, travel), variations in sample sizes or how rurality was defined across different studies [8]. For example, our findings and several others are based on self-reported PA data [8,11], McCormack and Meendering, 2016, included self-reported and device-measured PA, while findings from McCrorie et al. [7,18] are device-based. These inconsistencies highlight the need for standardized PA measurement tools [33] that can be easily adapted to account for culture and contexts as well as operationally comparable definitions of rurality in studies that include both urban and rural participants.

Few studies have compared screen time between the youth from urban and rural communities. Two studies have reported no differences [34,35], one reported less screen time among rural adolescents compared to urban counterparts [8], and one found that only urban males, but not females were more likely than their rural counterparts to have high video and computer use [36]. Another study comparing sedentary behavior between urban versus rural Canadian and American youth reported mixed results [37]. Youth from rural communities in the USA had higher odds of high television use but lower odds of high computer use compared to their urban counterparts; the opposite was found among Canadian youth [37]. Our study found that both male and female youth living in urban communities had lower odds of meeting the screen time recommendation than their rural counterparts. This aligns with findings from Christiana, Bouldin, and Battista (2021) in which rural youth had significantly less screen time than non-rural youth. As reflected in the mixed results reported by Carson et al. (2011), the contextual differences, as well as various options for leisure activities between urban and rural communities might explain the differences in their screen times. Regardless of location, a significant proportion of Canadian youth are not meeting the screen time recommendation. However, given that we observed that living in an urban community was associated with a reduced likelihood of meeting the screen time recommendation, future work should consider this important correlate. Given the preponderance of evidence linking high screen time to negative health outcomes [3,38,39], interventions and targeted messaging are needed to reduce screen time among youth.

The observed higher odds for males meeting the screen time recommendation in the springtime (April–June) may be related to the documented general decrease in sedentary time during spring compared to winter [40,41,42] or the start of outdoor sporting activities in which males tend be more active than females, thus, potentially displacing their sedentary time. This finding is also possibly related to the mediating/moderating or confounding effects of other unmeasured variables.

Both device-measured and self-reported data have previously shown significant sex differences in PA and screen time among youth. For example, data from the International Children’s Accelerometry Database (ICAD) show that females (2–18 years) were consistently less active than males [29]. Analyses of the Global School-based Student Health Survey among youth from 146 countries and territories [5] or 47 Latin American countries [30] demonstrated that more males meet the PA recommendation than females. Likewise, data collected over eight years for the Health Behavior in School-Aged Children (HBSC) study in 32 countries [28] showed similar results. These findings align with ours, showing that among Canadian youth, more males report meeting the PA recommendation than females. Sex/gender differences in PA among youth have been attributed to lower female participation in organized sport [43], gender roles [30], perceived barriers, perceived competence and different opportunities for independent mobility [44]. More equitable opportunities for participation in PA and interventions specifically targeted to promote PA for young females are needed. In the present study, a higher proportion of males did not meet the screen time recommendation than females which is consistent with results from previous studies [31,45,46]. Given that research suggests sex-specific associations between some health risks and PA [47] or screen time [48], as well as the gendered nature of these behaviors, it is important to continue including sex-disaggregated analyses in studies [48]. Furthermore, the findings reinforce the need to consider sex-specific requirements when planning related interventions and strategies to ensure equitable access to PA opportunities.

### Limitations and Strengths

Although nationally representative, the measures in the CCHS are self-reported and therefore our results are prone to social desirability and recall biases. However, self-reported PA from the CCHS has acceptable reliability [24] and population-level estimates align with accelerometer measured PA data [25]. Our analyses did not explore and tease out the nuances between domain-specific PA. The CCHS is a cross-sectional study which limits our ability to establish causal relationships. Therefore, further investigation using longitudinal data is required to build on our findings. Although not significantly different on most covariates to those with complete data, excluding respondents with missing PA and screen time data could have potentially biased our findings. Additionally, because we used a dichotomous definition of rurality, it is possible that we might have missed important contextual information which we would potentially have had, had we used a multi-level (e.g., four or seven level) variable corresponding to census metropolitan influence zones. However, previous analyses for adult data of the CCHS indicated that either variable captured similar patterns for odds of meeting the PA recommendation [49]. We were also unable to analyze combined 24 h movement behavior data, i.e., meeting sleep, PA and screen time recommendations combined, because sleep data were not available. Despite these limitations, our study has important strengths. First, in this study we performed sex-stratified analyses (often missing), examined both PA and screen time and controlled for important covariates. Furthermore, our findings shed light on the need to consider context (e.g., urban vs. rural) in studies and may lead to separate and better designed interventions and priority areas.

## 5. Conclusions

While significant proportions of Canadian youth do not meet PA and screen time recommendations, our study shows that living in an urban compared to a rural community may have a different impact on these behaviors. Additionally, the associations between PA and geographical location appear to be sex dependent. Future research should explore reasons for these differences and whether the associations are causal.

## Figures and Tables

**Table 1 ijerph-19-04394-t001:** Survey-weighted descriptive characteristics for urban versus rural participants.

Variable	Urban (*n* = 4390)	Rural (*n* = 1786)
	% (95% CI)	% (95% CI)
Age group		
12–13 years	30.9 (29.9, 32.0)	33.0 (31.1, 34.9)
14–15 years	33.7 (32.7, 34.8)	33.8 (32.2, 35.3)
16–17 years	35.3 (35.3, 35.4)	33.2 (32.8, 33.5) *
BMI category		
Thinness	2.7 (2.3, 3.1)	1.8 (0.9, 2.6)
Normal weight	72.8 (70.6, 75.0)	70.3 (67.5, 73.1)
Overweight	17.2 (16.5, 17.9)	18.0 (16.6, 19.4)
Obese	7.4 (6.3, 8.5)	9.9 (9.4, 10.5) *
Sex		
Female	48.2 (47.5, 48.9)	52.2 (47.7, 54.7)
Racial background		
White	63.7 (61.2, 66.2)	94.3 (94.1, 94.4) *
Season		
January–March	26.0 (24.8, 27.1)	23.0 (22.4, 23.6) *
April–June	26.4 (25.5, 27.4)	25.6 (23.0, 28.2)
July–September	18.8 (17.3, 20.3)	22.1 (17.6, 26.7)
October–December	28.8 (27.2, 30.5)	29.3 (27.9, 30.7)
Annual household income		
CAD 0–29,999	10.4 (9.8, 11.1)	6.9 (6.0, 7.8) *
CAD 30,000–59,999	18.0 (17.8, 18.3)	16.0 (14.1, 17.9)
CAD 60,000–99,999	23.5 (21.7, 25.4)	26.1 (22.9, 29.3)
CAD 100,000–149,999	21.7 (21.3, 22.1)	26.1 (25.9, 26.3) *
CAD 150,000+	26.3 (24.9, 27.7)	24.8 (20.9, 28.8)
Perceived general health		
Very good/excellent	75.4 (75.0, 75.9)	76.8 (76.8, 76.9) *
Perceived mental health		
Very good/excellent	75.0 (74.9, 75.1)	77.7 (76.1, 79.4) *
Met recommendation		
Physical activity	59.1 (58.7, 59.6)	58.7 (57.2, 60.2)
Screen time	32.1 (30.2, 34.0)	41.4 (40.1, 42.8) *
Physical activity and screen time	22.0 (19.5, 24.5)	26.9 (26.5, 27.3) *

* Rural respondents are significantly different from urban at *p* < 0.05; BMI = body mass index; Meeting physical activity recommendation = an average of 60 min per day of total moderate-to-vigorous intensity PA; meeting screen time recommendation = ≤2 h per day of recreational screen time.

**Table 2 ijerph-19-04394-t002:** Odds of meeting the PA recommendation between urban and rural Canadian youth by sex.

Variables	Model 1		Model 2		Model 3	
	Male	Female	Male	Female	Male	Female
Rural	1.00	1.00	1.00	1.00	1.00	1.00
Urban	0.93 (0.83, 1.05)	1.07 (0.95, 1.21)	1.00 (0.82, 1.20)	1.03 (1.02, 1.04) *	1.01 (0.86, 1.18)	1.05 (0.99, 1.11)
**Individual Variables**						
Age group						
12–13 years			1.00	1.00	1.00	1.00
14–15 years			1.38 (1.22, 1.55) *	1.06 (0.83, 1.36)	1.40 (1.38, 1.43) *	1.08 (0.85, 1.37)
16–17 years			1.31 (1.22, 1.42) *	0.85 (0.55, 1.30)	1.30 (1.13, 1.49) *	0.81 (0.59, 1.10)
BMI category						
Thinness			0.47 (0.24, 0.91) *	0.48 (0.34, 0.66) *	0.47 (0.23, 0.93) *	0.46 (0.32, 0.65) *
Normal weight			1.00	1.00	1.00	1.00
Overweight			0.97 (0.75, 1.27)	0.73 (0.68, 0.79) *	0.98 (0.79, 1.23)	0.82 (0.74, 0.92) *
Obese			0.77 (0.64, 0.92) *	0.67 (0.67, 0.68) *	0.78 (0.67, 0.90) *	0.76 (0.66, 0.87) *
Perceived general health						
Poor/fair/good			1.00	1.00	1.00	1.00
Very good/excellent			1.64 (1.30, 2.06) *	1.07 (0.88, 1.29)	1.76 (1.60, 1.93) *	1.10 (0.77, 1.56)
Perceived mental health						
Poor/fair/good			1.00	1.00	1.00	1.00
Very good/excellent			1.40 (1.38, 1.42) *	1.34 (1.23, 1.45) *	1.30 (1.06, 1.60) *	1.25 (1.04, 1.51) *
**Annual Household Income**						
CAD 0–29,999					0.72 (0.69, 0.74) *	0.34 (0.31, 0.36) *
CAD 30,000–59,999					0.86 (0.80, 0.93) *	0.48 (0.48, 0.49) *
CAD 60,000–99,999					0.88 (0.72, 1.07)	0.66 (0.56, 0.78) *
CAD 100,000–149,999					0.95 (0.72, 1.26)	0.86 (0.81, 0.91) *
CAD 150,000+					1.00	1.00
**Data Collection Season**						
January–March					1.00	1.00
April–June					1.50 (1.39, 1.62) *	1.19 (0.85, 1.68)
July–September					0.94 (0.78, 1.12)	1.21 (0.74, 1.97)
October–December					1.08 (0.96, 1.21)	1.07 (0.88, 1.30)

* Statistically significant difference at *p* < 0.05; BMI = body mass index.

**Table 3 ijerph-19-04394-t003:** Odds of meeting the screen time recommendation between urban and rural Canadian youth by sex.

Variables	Model 1		Model 2		Model 3	
	Male	Female	Male	Female	Male	Female
Rural	1.00	1.00	1.00	1.00	1.00	1.00
Urban	0.64 (0.53, 0.78) *	0.70 (0.65, 0.75) *	0.70 (0.69, 0.70) *	0.74 (0.59, 0.92) *	0.71 (0.65, 0.77) *	0.71 (0.59, 0.86) *
**Individual Variables**						
Age group						
12–13 years			1.00	1.00	1.00	1.00
14–15 years			0.48 (0.32, 0.72) *	0.51 (0.43, 0.59) *	0.48 (0.30, 0.78) *	0.47 (0.44, 0.49) *
16–17 years			0.48 (0.44, 0.53) *	0.53 (0.48, 0.59) *	0.49 (0.44, 0.54) *	0.51 (0.41, 0.62) *
BMI category						
Thinness			0.72 (0.38, 1.36)	1.21 (1.08, 1.36) *	0.74 (0.40, 1.36)	1.08 (0.98, 1.20)
Normal weight			1.00	1.00	1.00	1.00
Overweight			0.94 (0.52, 1.69)	0.77 (0.62, 0.95) *	0.95 (0.50, 1.83)	0.72 (0.45, 1.14)
Obese			0.65 (0.31, 1.36)	0.49 (0.34, 0.70) *	0.66 (0.31, 1.44)	0.56 (0.35, 0.88) *
Perceived general health						
Poor/fair/good			1.00	1.00	1.00	1.00
Very good/excellent			1.8 (1.09, 2.96) *	1.58 (1.25, 1.99) *	1.82 (1.03, 3.21) *	1.50 (1.07, 2.09) *
Perceived mental health						
Poor/fair/good			1.00	1.00	1.00	1.00
Very good/excellent			1.09 (0.74, 1.63)	1.41 (1.36, 1.46) *	1.04 (0.76, 1.42)	1.37 (1.29, 1.46) *
**Annual Household Income**						
CAD 0–29,999					0.57 (0.34, 0.96) *	0.76 (0.74, 0.78) *
CAD 30,000–59,999					0.87 (0.60, 1.26)	0.41 (0.40, 0.41) *
CAD 60,000–99,999					0.80 (0.80, 0.81) *	0.55 (0.36, 0.83) *
CAD 100,000–149,999					0.72 (0.56, 0.92) *	0.70 (0.58, 0.83) *
CAD 150,000+					1.00	1.00
**Data Collection Season**						
January–March					1.00	1.00
April–June					1.33 (1.18, 1.49) *	0.93 (0.84, 1.03)
July–September					1.03 (0.61, 1.72)	1.21 (0.78, 1.88)
October–December					0.95 (0.88, 1.02)	1.41 (1.26, 1.58) *

* Statistically significant difference at *p* < 0.05; BMI = body mass index.

**Table 4 ijerph-19-04394-t004:** Odds of meeting both PA and screen time recommendations between urban and rural Canadian youth by sex.

Variables	Model 1		Model 2		Model 3	
	Male	Female	Male	Female	Male	Female
Rural	1.00	1.00	1.00	1.00	1.00	1.00
Urban	0.71 (0.62, 0.81) *	0.83 (0.73, 0.95) *	0.74 (0.73, 0.76) *	0.80 (0.56, 1.16)	0.75 (0.71, 0.81) *	0.82 (0.58, 1.15)
**Individual Variables**						
Age group						
12–13 years			1.00	1.00	1.00	1.00
14–15 years			0.63 (0.47, 0.86) *	0.54 (0.48, 0.61) *	0.64 (0.43, 0.94) *	0.51 (0.45, 0.58) *
16–17 years			0.61 (0.56, 0.66) *	0.57 (0.47, 0.70) *	0.62 (0.59, 0.64) *	0.53 (0.50, 0.55) *
BMI category						
Thinness			0.85 (0.33, 2.20)	0.91 (0.78, 1.06)	0.88 (0.33, 2.34)	0.81 (0.66, 1.00)
Normal weight			1.00	1.00	1.00	1.00
Overweight			0.93 (0.59, 1.45)	0.70 (0.52, 0.94) *	0.94 (0.58, 1.53)	0.68 (0.41, 1.13)
Obese			0.68 (0.48, 0.96) *	0.38 (0.37, 0.40) *	0.68 (0.46, 1.02)	0.45 (0.34, 0.60) *
Perceived general health						
Poor/fair/good			1.00	1.00	1.00	1.00
Very good/excellent			1.92 (1.30, 2.82) *	1.43 (1.28, 1.60) *	1.96 (1.21, 3.16) *	1.39 (1.33, 1.46) *
Perceived mental health						
Poor/fair/good			1.00	1.00	1.00	1.00
Very good/excellent			1.18 (0.86, 1.62)	1.69 (1.65, 1.74) *	1.11 (0.91, 1.36)	1.54 (1.26, 1.89) *
**Annual Household Income**						
CAD 0–29,999					0.57 (0.45, 0.73) *	0.55 (0.30, 1.03)
CAD 30,000–59,999					0.90 (0.67, 1.22)	0.35 (0.29, 0.43) *
CAD 60,000–99,999					0.78 (0.76, 0.79) *	0.54 (0.32, 0.89) *
CAD 100,000–149,999					0.71 (0.62, 0.80) *	0.88 (0.87, 0.90) *
CAD 150,000+					1.00	1.00
**Data Collection Season**						
January–March					1.00	1.00
April–June					1.22 (1.17, 1.26) *	0.99 (0.76, 1.30)
July–September					0.85 (0.45, 1.62)	1.33 (0.61, 2.89)
October–December					0.86 (0.79, 0.94) *	1.27 (1.00, 1.60) *

* Statistically significant difference at *p* < 0.05; BMI = body mass index.

## Data Availability

The dataset used and/or analyzed for the present study is available to researchers pending proposal approval through the Research Data Centres (RDC) Program at Statistics Canada (https://www.statcan.gc.ca/eng/rdc/index, accessed on 2 April 2022).

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
