# Peer review of "A Comparison of Meeting Physical Activity and Screen Time Recommendations between Canadian Youth Living in Rural and Urban Communities: A Nationally Representative Cross-Sectional Analysis"

_ijerph, 2022, doi:10.3390/ijerph19074394_

Round 1

Reviewer 1 Report

The study is interesting and presents a result that can guide actions to promote physical activity in urban and rural contexts, based on the specificities of each one. I believe that the authors can indicate paths, in the discussion, for further research considering the specificities of the urban and rural settings, not only pointing out that longitudinal studies are necessary.

One point I would like to highlight is that the authors could divide the paragraphs more, as they are long and prevent a more fluid reading. Perhaps it could be an issue of writing style.

Considering their presence in methodological quality assessment instruments, I question whether the person who conducted the analysis was blinded to the exposures and outcome. This is a point that seems very important to me.

I suggest that authors should be able to answer the items on the STROBE checklist in order to improve the quality of reporting.

I believe that the self-report questions (questionnaire) and the design are not limiting for the study. In my assessment, they are inherent limitations of the measure and the design. But on the other hand, I reflect that accelerometers and other study designs also have important limitations. Thus, I suggest that the limitations be more focused on the definition of rurality and how much they may have influenced the final result/conclusions.

Author Response

Please find attached herewith, our response to Reviewer 1's comments

Reviewer 2 Report

Summary

The authors reported on the prevalence of rural vs. urban Canadian youth (aged 12-17 years) meeting physical activity and screen time recommendations as defined by the Canadian 24-Hour Movement Guidelines. Further, the authors stratified their results by sex and reported odds ratios of meeting guidelines by age group, BMI category, perceived general health, perceived mental health, annual household income, and data collection season. This article provided up to date information on the prevalence and likelihood of different Canadian youths meeting physical activity and screen time recommendations. 

General comments

The authors did a good job identifying the most recent guidelines (Canadian 24-Hour Movement Guidelines) for youth physical activity or screen time recommendations and providing a clear, thorough analysis on how different Canadian youth may spend their time regarding physical activity and screen time. However, the Candian 24-Hour Movement Guidelines are meant to be practised and investigated in combination.  

“This fundamental shift from focusing on movement behaviours in isolation to the concept that “the whole day matters” is strongly supported by the available evidence. Consideration of all behaviours along the movement continuum as a collective is warranted, and holds promise in the promotion of population health.”(Tremblay et al., 2016)

Therefore, there are two major points the authors’ should consider: 

  1. Analyses of meeting/not meeting both physical activity and screen time recommendations should be conducted. Exploring if the recommendations are met in combination will give us better insight into how girls and boys from rural vs. urban Canada differ. For example, are the same children who don’t meet the physical activity recommendation also not meeting screen time recommendations? If so, what are the implications of this? 
  2. If possible, youth meeting/not meeting sleep recommendations should be added to the analysis. If this is not possible, it should be acknowledged as a limitation. 

The variables in the methods can be elaborate on: 

  1. For physical activity, what was the rationale for using a daily average of 60 minutes MVPA/day versus other measurements (e.g., how many days 60 min MVPA/day were met)? 
  2. For screen time, the authors reported daily average screen time was generated for week and weekend days. Was a weighted average used? 
  3. Using >150,000 for the reference category for annual household income may provide a clearer picture of how children from different socioeconomic backgrounds meet recommendations. The data now makes it look like income has a big effect across all categories, but it’s mainly those from the lowest income bracket that is different from everyone else. Please update this or provide a rationale and discussion regarding your findings related to income. 

Specific comments 

In the abstract, on line 31, the authors state that “Urban females had significantly higher odds of meeting the PA recommendation”. Meeting the PA recommendation was not significantly higher, but the odds of meeting the PA recommendation were “statistically significant”.  

On lines 91-95 of the introduction the authors mention, “Furthermore, current evidence (van Sluijs et al. 2021) and trends suggest that the prevalence of insufficient PA and high screen time among youth may, at best be stable (Gelez-Nobrega et al. 2020) or increasing (Guthold et al. 2020) over time, thus, it is important to understand if the previously observed differences in self-reported PA adherence between urban and rural youth remain or have increased.” This is an important and relevant point the authors bring up. However, in measuring rurality, the authors used a different definition to define urban and rural locations than previous studies that measured physical activity prevalence on earlier cycles of the same dataset. (Refer to lines 291-302 of the discussion)

Would it be possible to use the same definition of rurality to ensure consistency and better compare the results? If not, please provide a rationale for why you chose to use a different measurement. 

In the methods (lines 160-162), the seasons of data collection are split up as January-March; April-June; July-September; October-December. What is the rationale for why the months of the seasons were split up in this manner? Further, the results suggest that boys have higher odds of meetings screen time guidelines in April-June, but for girls it’s July-December. Why might this be? Please add your explanation to the discussion. 

A similar pattern to the seasons is seen with income. Boys from higher-income families have higher odds of meeting the screen time recommendations while the opposite is true for girls. Please discuss.

Author Response

Please herewith attached our responses to reviewer 2's comments

Reviewer 3 Report

SUMMARY

The aim of the present study was to determine the relationship between the leaving environment (urban vs rural) and the fact of meeting or not the physical activity and screen time recommendations.

The main contribution of this study is to provide an updated view of the self-reported physical activity and screen time behaviors of young people in Canada.

GENERAL COMMENTS

The introduction section was very informative, with a good effort from the authors to explain what is the originality and the interest of the study.

The study has several limits evocated by the authors: self-reported methods, incomplete data for the last year of the considered period of measurement (2017-2019), the use of a dichotomous definition of rurality.

Regarding more precisely the writing of the manuscript, I have had some difficulties to fully understand what statistical analyses were actually performed, and I am not convinced that all presented analyses are appropriate, as explained in my specific comments below.

SPECIFIC COMMENTS

  1. Please provide the reference of the physical activity questionnaire and information about the validity if any.
  2. Lines 159-161: It could be easier for the reader to precise that numbers for BMI are actually z-scores.
  3. I have found the sentence “For descriptive statistics, statistically significant differences were identified if the 95% confidence intervals did not overlap.” (lines 174-175) unclear. What were the groups to be compared ? Rural vs urban? Anyway, there are two issues with this analysis:
    1. I do not think that this procedure is fully appropriate to claim significance because in principle, one does not need to have non-overlapping 95% confidence intervals to have a significant difference at an alpha of 0.05. This is why the graphical method that uses confidence intervals for the two scores to be compared is hard to interpret. It is easier to interpret the 95% confidence interval of the difference between the compared scores and to investigate if that interval crosses 0 or not. If the interval crosses 0, here we could argue for a significant difference at an alpha of 0.05. If significance has to be detected, why could the authors not conduct statistical tests for differences of proportions or means depending on the variables to be compared? Surprisingly, it seems the authors conducted this kind of analysis when we read the legend of table 1. Thus, this should be indicated in the statistical part of the Methods section if these analyses are demonstrated as useful. Moreover, as several comparisons tests are performed, the authors should control the false positive error rate that rises during multiple testing.
    2. I am wondering what was the interest/value of this analysis in addition to the results provided by the logistic model.
  4. Regarding the logistic model, how did the authors manage the fact that the 2019 cycle has only three counties (from a statistical point of view)?
  5. Lines 201-202: It is surprising not to show the results of the interaction term while this may have a substantial consequence on the results and their presentation by the authors.
  6. Lines 209-210 (“Table 3 presents results for females, which found statistically significantly greater odds of meeting the PA recommendation among females living in rural compared to urban communities (OR = 1.04, 95% CI: 1.02 - 1.05”): The authors could indicate that this result is related to the more adjusted model (models 3 and 4).
  7. Discussion section, lines 349-350, “In the present study, males had higher screen times than females”: I have not seen where was this information. The authors should ensure that all statements are based on data that are actually shown in the manuscript.

Author Response

Please herewith attached our responses to reviewer 3's comments

Round 2

Reviewer 2 Report

Dear Authors, 

Thank you very much for re-submitting your paper and addressing the comments I have provided. You did a thorough job responding to my comments and provided clear explanations about why certain suggestions could not be addressed. 

I feel this paper is now suitable for publication. Well done!

Author Response

We would like to thank the reviewer for their diligence and thorough review. We appreciated their comments and suggestions which we believe have only strengthened our paper. We are glad that we have sufficiently responded to the reviewer's comments.

Reviewer 3 Report

I thank the authors for their answers and explanations. I have still some minor comments:

  1. The authors should provide the reference regarding the validity of the physical activity (PA) questionnaire assessed against accelerometer (Colley et al., 2019) in the Methods section (2.3.1). I think it is better to know before reading the results that the PA questionnaire has previously been studied.
  2. I still have a remark regarding my comment about statistical analyses (cf. point 3 from my previous review). One could think it is unfortunate not to be able to conduct desired analyses “just” due to technical (software) limitations. However, because it does not concern the main analysis in relation to the main objective of the paper, one could consider the author’s approach as acceptable (I precise that I am dealing with the issue of using non-overlapping 95% CIs without adjustment for multiple comparisons to compare the proportions (rural vs urban)). The authors could inform the reader in the statistical section that using non-overlapping 95% CIs is a more conservative approach of claiming significance at a 5% alpha level.
  3. Regarding my point 4) from my previous review, I totally agree that table 1 is of interest. But one can provide descriptive statistics while not having to show inferential statistics if the role of those inferential statistics is not to mainly support the main study objective. The use of the CIs for proportions (table 1) in the present paper may be fine as long as the authors do not use them to mainly support their conclusions, which seems to be the case here.
  4. In my point 8) from my previous review, I thought the authors did not provide information in the results section about the sentence “In the present study, males had higher screen times than females” (lines 320-321, new version of the manuscript). I realize that this kind of formulation (that from the authors) could be misleading and could make harder than necessary the understanding of the results. With the expression “higher screen time”, one could consider screen time as a continuous variable and expect to see in the results section the time (in minutes) actually spent with a screen. However, time information do not appear as continuous variables in the paper, but as proportions of meeting recommendations (this is the same for physical activity). The authors use the expression “higher screen times” for males to describe the fact that the proportion of men meeting the screen time recommendations is smaller than for females. If indeed it seems logical that not meeting screen time recommendations is likely related to higher screen time at the group level, I suggest trying to use results descriptions that are more consistent with the kind of data that is described.

Author Response

We thank the reviewer for their comments. We have modified our manuscript as suggested by the reviewer and have responded to their comments in the attached document. 
